# Chronic Exposure to Paraquat Induces Alpha-Synuclein Pathogenic Modifications in *Drosophila*

**DOI:** 10.3390/ijms222111613

**Published:** 2021-10-27

**Authors:** Jean-Noël Arsac, Marianne Sedru, Mireille Dartiguelongue, Johann Vulin, Nathalie Davoust, Thierry Baron, Bertrand Mollereau

**Affiliations:** 1French Agency for Food, Environmental and Occupational Health & Safety (Anses) Laboratory of Lyon, Neurodegenerative Diseases Unit, University of Lyon, F-69342 Lyon, France; jean-noel.arsac@anses.fr (J.-N.A.); mireille.dartiguelongue@gmail.com (M.D.); johann.vulin@anses.fr (J.V.); 2Laboratory of Biology and Modelling of the Cell, UMR5239 CNRS/ENS de Lyon, INSERM U1210, UMS 3444 Biosciences Lyon Gerland, University of Lyon, F-69342 Lyon, France; marianne.sedru@gmail.com (M.S.); nathalie.davoust-nataf@ens-lyon.fr (N.D.)

**Keywords:** Parkinson’s disease, alpha-synuclein, *Drosophila*, paraquat

## Abstract

Parkinson’s disease (PD) is characterized by the progressive accumulation of neuronal intracellular aggregates largely composed of alpha-Synuclein (αSyn) protein. The process of αSyn aggregation is induced during aging and enhanced by environmental stresses, such as the exposure to pesticides. Paraquat (PQ) is an herbicide which has been widely used in agriculture and associated with PD. PQ is known to cause an increased oxidative stress in exposed individuals but the consequences of such stress on αSyn conformation remains poorly understood. To study αSyn pathogenic modifications in response to PQ, we exposed *Drosophila* expressing human αSyn to a chronic PQ protocol. We first showed that PQ exposure and αSyn expression synergistically induced fly mortality. The exposure to PQ was also associated with increased levels of total and phosphorylated forms of αSyn in the *Drosophila* brain. Interestingly, PQ increased the detection of soluble αSyn in highly denaturating buffer but did not increase αSyn resistance to proteinase K digestion. These results suggest that PQ induces the accumulation of toxic soluble and misfolded forms of αSyn but that these toxic forms do not form fibrils or aggregates that are detected by the proteinase K assay. Collectively, our results demonstrate that *Drosophila* can be used to study the effect of PQ or other environmental neurotoxins on αSyn driven pathology.

## 1. Introduction

Parkinson’s disease (PD) is characterized by the neuronal accumulation of αSyn-rich cytoplasmic inclusions known as Lewy bodies [1]. αSyn is a vertebrate-specific 140 amino-acid presynaptic protein, which can acquire neurotoxic properties during aging and in synucleinopathies such as PD [2]. During this process, soluble αSyn monomers form oligomers that progressively accumulate to produce large αSyn fibrils that constitute the insoluble aggregates [3]. Interestingly, αSyn aggregation is associated with increased αSyn phosphorylation at Ser 129 (αSynp129), which has been identified as the predominant pathogenic modification of αSyn present in Lewy bodies in PD and other synucleinopathies [4]. The accumulation of misfolded αSyn, under the form of fibrils and/or aggregates, was shown using the resistance of αSyn to proteinase K digestion assay in cellular protein extracts from PD and related disorders patient [5,6,7,8].

There are multiple factors that can promote or enhance the aggregative process. This includes familial missense mutations of αSyn, such as αSynA53T, which are associated with an increased aggregation propensity [9]. Abnormal elevated expression of αSyn also promotes its aggregation. This is observed in familial duplication or triplication of the synuclein alpha (SNCA) gene, as well as in a risk variant in the non-coding distal element of SNCA, which all lead to a constitutive increased αSyn expression [3,10].

The study of PD pathogenesis strongly relies on animal models, in which the appearance of PD hallmarks can be followed, allowing the identification of genetics factors regulating their formation and progression. Although non-human primates most closely mimic the human pathology, mice are the most often used animal model [11]. The neuronal overexpression of human αSyn in mouse neurons can be associated with αSyn aggregation and locomotor impairment although the loss of dopaminergic neurons is often not observed [12]. For example, it was shown that misfolded and hyperphosphorylated αSyn that are resistant to proteinase K digestion are found in transgenic mice with motor deterioration [6]. One drawback with transgenic mouse expressing the human SNCA gene is that the endogenous mouse α- and β-synucleins interfere with human αSyn fibril formation and appearance of Lewy bodies/neurites, thus limiting a full understanding of the aggregative process in these mice [13].

Another valuable model of Parkinson is *Drosophila melanogaster*, or the fruit fly, to study genetic and environmental factors involved in PD [14,15,16]. The fly is indeed a powerful genetic model organism that carries a complex neuronal circuitry including dopaminergic neurons clusters that express dopamine and evolutionary conserved enzymes, such as the tyrosine hydroxylase [17,18]. Several models of PD were developed in *Drosophila*, including a well-characterized model in which the neuronal overexpression of human SNCA gene using the UAS/GAL4 system mimics PD hallmarks [19]. This model was greatly used to study and characterize conserved cellular mechanisms associated with αSyn pathology in humans, despite the fact that *Drosophila* does not carry an endogenous ortholog of SNCA gene [12,20,21]. The common parkinsonian hallmarks observed in flies overexpressing SNCA include the accumulation of αSyn aggregates in situ, loss of dopaminergic neurons (DA), and the impairment of fly locomotion (negative geotaxism) [17,19,22,23,24]. The in situ detection of αSyn aggregates, and the neurodegeneration were best observed with UAS-αSyn transgenic insertion expressing a fly codon optimized form of αSyn or insertions using the QF/QUAS system that all result in a high expression level of αSyn [25,26]. Although useful, these high-expressing αSyn lines exhibiting high level of neurodegeneration, are not optimized to study the progressive and pathogenic accumulation of αSyn during aging. Flies expressing SNCA with UAS/GAL4 at a more moderate level were previously used to follow αSyn pathogenic modifications, including the detection of phosphorylated αSynp129 and the accumulation of misfolded αSyn resistant to mild-digestion by proteinase K during aging [22,27]. However, in these models the detection of αSyn aggregates in situ was best observed by co-expressing an additional aggregate prone proteins, such as Huntingtin or Tau proteins [28,29]. Collectively, these results show that *Drosophila* can be used to study αSyn pathogenic modification although the detection of αSyn aggregates and inclusions remains quite challenging during fly aging. One goal of this study was thus to optimize the available labeling methods of total αSyn and αSynp129 to follow αSyn pathogenic modifications in *Drosophila* adult brains during aging.

Environmental stresses, such as the exposure to pesticides including herbicides, have been associated with PD. Indeed, pesticides targeting the mitochondria, including PQ, 1-methyl-4-phenyl-1,2,3,6-tetrahydropyridine (MPTP) and rotenone provoke PD in humans and parkinsonian symptoms in animal models of PD [30]. Furthermore, the herbicide, PQ, has been widely used in agriculture and associated with PD contracted by farmers and winegrowers/occupational exposure [31,32]. PQ exposure on recombinant αSyn was also shown to enhance the rate of αSyn fibril formation in vitro [33]. In the same study, PQ oral exposure was associated with αSyn increased expression and aggregate formation in mice. However, it is not clear whether PQ simply promotes all or part of αSyn pathogenic modifications, including increased αSyn levels and phosphorylation, αSyn misfolding and aggregation, that are observed in PD.

PQ exposure is also commonly used to study PD in *Drosophila* that do not overexpress human SNCA gene [20,34,35,36]. In this model, fly mortality was used as proxy of PQ toxicity to identify the mechanism regulating the stress response to PQ [15,37,38]. In some studies, but not all, PQ acute exposure was also associated with the loss of dopaminergic neurons [37,39,40]. However, the analysis of the effect of PQ on αSyn pathogenic modifications was not assessed in these studies. In the present study, we established the use of PQ in a chronic manner to model PD in flies overexpressing human αSyn in order to analyze the PQ-induced αSyn pathogenic modifications. To achieve this goal, we first used in situ detection and biochemical assays previously used to follow αSyn pathogenic modifications [5,41,42,43,44,45,46,47,48,49,50] (Table 1). These biochemical assays include the detection of various status of aggregation of αSyn in response to treatment with denaturating buffers of increasing strength and in the resistance to proteinase-K digestion assay in aging flies. We next established a PQ chronic exposure protocol on flies expressing αSynWT or αSynA53T, in which we followed fly viability and αSyn pathogenic modifications. We found that PQ-induced fly mortality was enhanced by the expression of αSynWT or αSynA53T. The exposure to PQ was also associated with increased levels of total αSyn and αSynp129 in the subesophageal ganglion and the antennal lobes. Finally, we showed that PQ promoted the accumulation of αSyn resistant to mild denaturating buffers but not to proteinase-K digestion, suggesting that PQ rather induced the accumulation toxic soluble misfolded forms of αSyn, that yet do not form fibrils and aggregates. Collectively, our results demonstrate that the chronic exposure of PQ in fly expressing human αSyn can be used to study the effect of environmental neurotoxins on αSyn pathology.

## 2. Results

### 2.1. Developing Immuno-Histological Labeling to Follow αSyn Pathology during Aging in Drosophila

The immunolabeling of αSyn protein and its forming aggregates is a critical aspect of the characterization of PD pathology. αSyn aggregates have been previously reported in neurons of aged transgenic *Drosophila* expressing αSyn [22,26]. However, depending on the level of the expression of αSyn, the detection of αSyn aggregates is not routinely reported due to issues in their detection in tissues of *Drosophila* transgenic lines [20,21,25,28,29]. To improve the method of αSyn immunodetection in situ, we examined αSyn labeling in dissected *Drosophila* brains stained with three commonly used anti-αSyn antibodies (Syn 211, MJFR1, and C20R), which recognize different epitopes in the C-terminus of the protein. We labeled αSyn protein by immunofluorescence in brains of young (5–10 days) and old flies (60 days) expressing the wild type form of human αSyn (αSynWT) or the A53T variant (αSynA53T) under the control of the pan neuronal driver ELAV (Figure 1). The stainings appeared globally stronger in antennal lobes, suboesophageal ganglions and in mushroom bodies, compared to the retina and other neuropils in young flies and to a lesser extent in old flies (Figure 1a–l). To precisely evaluate αSyn detection in young and old flies, we quantified the immunofluorescent intensity in the antennal lobes, which can easily be delineated, and in neuropils of the anterior ventrolateral protocerebrum using the three anti-αSyn antibodies (Figure 1m,n). We observed a more intense immunofluorescent αSyn staining in young flies expressing αSynWT or αSynA53T compared with older flies of the same genotype with the Syn 211 antibody (Figure 1a,d,g,j,m,n). Interestingly, for the MJFR1 antibody, the loss of αSyn labeling in *Drosophila* brain during aging is less obvious than with Syn 211 in the antennal lobes for αSynA53T and is not observed in neuropils of the anterior ventrolateral protocerebrum neuropils in flies expressing αSynWT or αSynA53T (Figure 1m,n). In contrast to Syn 211 and MJFR1 antibodies, the labeling of αSynWT and αSynA53T with the C20R antibody was still visible in the antennal lobes and was associated with a weak and uniform increase in the signal of αSyn labeling in the whole neuropil (Figure 1f,l,m,n). This may be due to the fact that the C20R antibody readily detects αSyn forming aggregates, as shown by the detection of both SDS-resistant oligomers in Western blot and specific ELISA immunoreactivity of such aggregates produced in vitro by Protein Misfolding Cyclic Amplification [51]. Collectively, these results demonstrate variability in the immunodetection of αSyn labeling in *Drosophila* brain during aging. This raises the possibility that αSyn epitopes detected by Syn 211 (111–125 amino acid sequence) and to a lesser extent MJFR1 (118–123 amino acid sequence) but not C20R (C-terminal unknown amino acid sequence) antibodies could be masked due to the change of αSyn conformation during aging. To test this hypothesis, we used an antigenic unmasking protocol adapted to immunohistochemistry with the MJFR1 antibody to detect the presence αSyn (see Material and Method section). Using this protocol and in contrast to the immunofluorescent labeling, we detected a strong αSyn immunoreactivity, especially in the subesophageal ganglion, the antennal lobes, the mushroom bodies and neuropil of both young and old flies with pan-neuronal expression of αSynWT or αSynA53T (Figure 1o–r). Interestingly, we also observed a global increased accumulation of αSyn by immunochemistry in subesophageal ganglion and neuropil of old compared to young flies (Figure 1p,r). Both the increased detection of αSyn using the unmasking protocol and the variability in the immunodetection of αSyn by immunofluorescence, suggest that αSyn accumulate in aggregated forms in the brain of aging flies.

### 2.2. Developing Biochemical Assays to Follow αSyn Pathology during Aging in Drosophila

Previous studies have also shown that the process of αSyn aggregation is associated with the accumulation of αSyn fibrils that are resistant to mild proteolysis by Proteinase K [49]. The accumulation of proteinase-K resistant forms of αSyn was also observed in flies carrying a deficiency in the *Drosophila* homolog of the glucocerebrosidase 1 (GBA1) gene [27]. Therefore, we examined whether αSyn resistance to proteinase-K digestion varied during fly aging. We found an age-dependent increased accumulation of proteinase-K resistant forms of αSyn in both transgenic *Drosophila* expressing αSynWT and αSynA53T with the ELAV driver (Figure 2a,b and Appendix A). We also observed that the sensitivity to proteinase-K digestion was higher in *Drosophila* expressing αSynWT compared to the one expressing αSynA53T, suggesting that αSyn protein carrying mutation A53T may be more prone to aggregate. The concentration of proteinase-K for which αSyn was degraded ranged from 0.5 to 1 μg/mL of proteinase-K and 1 to 1.5 μg/mL of proteinase-K in young (1-day-old) and old flies (40-day-old) expressing αSynWT or αSynA53T, respectively (Figure 2a).

We next determined the various stages of aggregation of αSynA53T and αSynWT by using extraction buffers of increasing denaturating strength (LB1 < LB2 < LB3). We noticed that while both αSynA53T and αSynWT were detected with similar levels in both LB1 and LB3 buffers in young flies, αSynA53T but not αSynWT was weakly detected after extraction with the weak denaturating buffer LB1 in old flies (Figure 2c,d). The proportion of αSynA53T quantity detected with LB3 relative to LB1, ranged from 47% to 73% in young and old flies, respectively (Figure 2d). The reduced detection in LB1 but higher detection in LB3 buffers suggests a progression of αSynA53T into the aggregative process in old flies. A more detailed characterization showed that αSynA53T was normally detected after protein extraction with LB1 in young (1-, 10-, 20-day-old) and 30-day-old flies but strongly decreased in 40-day-old and was totally undetectable in 50-day-old flies (Appendix A). A difference in αSynA53T detection was not observed with the buffers of higher denaturating strength (LB2 and LB3) (Appendix A).

Collectively these results show that αSyn detection after extraction in weakly denaturating buffer decreases while resistance to proteinase-K digestion progressively increases during fly aging. This further supports the proposal of an age-dependent accumulation of misfolded and aggregated forms of αSyn. Finally, the fact that αSynA53T exhibited a more pronounced resistance to proteinase-K digestion and to denaturation compared to αSynWT may be due to the fact that αSynA53T is more aggregate-prone than αSynWT [49] (see also discussion in Section 3).

To further analyze the progressive pathological alterations of αSyn protein in aging flies, we examined the presence of phosphorylation of αSyn at serine 129 (αSynp129), which was previously associated with the appearance of αSyn aggregates [4,22,42,52]. For that, we used an anti-αSynp129 (EP1536Y) antibody for the detection of αSynp129 by immunochemistry in paraffin-embedded head sections in young and old flies expressing αSynWT or αSynA53T with the ELAV driver (Figure 3). We observed that αSyn is constitutively phosphorylated at low levels especially in the suboesophageal ganglion area in both young αSynWT (Figure 3a,a′) and αSynA53T (Figure 3c,c′) expressing flies. We also noticed an increase of αSynp129 levels in these two conditions in old compared to young flies (Figure 3b,b′,d,d′,e,f). Collectively, our analysis provides a set of complementary methods to evaluate the levels of αSyn and the pathological modifications during fly aging including αSyn/αSynp129 detection by immunochemistry in situ, as well as resistance to proteinase-K digestion and solubility by immunoblotting.

### 2.3. Establishing a Chronic PQ Exposure Model in Drosophila Expressing Human αSyn

PQ exposure was used in *Drosophila* to study PD [20,34,35,36,38]. However, the acute dose of PQ used in these models (10 or 20 mM fed to adult flies, depending on the studies) led to rapid loss of fly viability within 2–3 days, which precluded a fine analysis of the effect of PQ on αSyn protein. A goal of this study was instead to further validate the use of a chronic exposure to PQ at a lower dose to determine whether αSyn pathogenic modifications resemble those observed in PD, such as increased expression and phosphorylation, increased resistance to proteinase-K digestion and to denaturating buffers.

To establish a model to study the progressive pathological modification of αSyn induced by PQ, we first measured the viability of flies expressing αSynWT, αSynA53T, or control GFP under the ELAV driver untreated or treated with PQ (Figure 4). The flies raised under a control diet (agarose + 10% sucrose) only (untreated) died at a similar rate but with an earlier onset in αSynWT (17 days) compared to αSynA53T and control GFP (19 days) conditions (Figure 4a). The fact that αSynWT, but not αSynA53T, exhibits a higher mortality than control flies is not clear, but we cannot exclude an effect due to the different genetic insertions of the P[UAS] transgenes. Using a chronic low-dose exposure PQ protocol (0.75 mM) during the entire length of the experiment (PQ treated), all transgenic flies succumbed to death before 23 days of age (Figure 4a); in all the conditions, PQ significantly enhanced the mortality rate. To further describe the impact of oral exposure to PQ, the results of toxicity were analyzed by monitoring medium lifespan and the slope steepness on survival curves. While the median lifespan significantly diminished in flies exposed to PQ for all transgenic lines, the reduction observed αSynA53T flies (49.73%) or αSynWT flies (43.36%) were significantly shorter than that for GFP (38%) (Figure 4b). Moreover, the linear regression between 75% survival and 15% survival flies for each experimental condition showed a steeper slope for αSynA53T flies exposed to PQ (−19.54), reflecting that flies die at higher rate in this condition compared with the others conditions (Figure 4c). We have thus established a chronic exposure paradigm to PQ, in which the PQ toxicity is synergistically enhanced by the expression of αSynWT or αSynA53T. In this paradigm, the lethal time 50 (LT50), at which 50% of the flies succumb from PQ treatment, is between 15 and 19 days, which allows sufficient time to accumulate pathogenic modifications on the αSyn protein. Therefore, LT50 will be the selected time point for the analysis of the effect of PQ on αSyn protein pathogenic modifications.

### 2.4. Chronic Exposure to PQ Leads to Pathogenic Modifications of α-Synuclein Protein in Drosophila

We and others have previously shown that PQ exposure increases the levels of αSyn in human dopaminergic neurons and in SH-SY5Y human neuroblastoma cell lines [53,54]. We thus first examined if the exposure to PQ affected the αSyn level using immunofluorescence and confocal detection with the anti-αSyn C20R antibody in flies expressing αSynWT (Figure 5a,b) or αSynA53T (Figure 5d,e). We observed a significant accumulation of αSyn in PQ exposed flies compared to untreated flies (Quantification in Figure 5c,f). We next examined the effect of PQ treatment on the status of aggregation of αSynA53T and αSynWT proteins by western blots by using extraction buffers of increasing denaturating strength (LB1 < LB3).

As observed on the immunoblots, we first noticed an increase of αSyn levels of expression in flies treated with PQ compared to untreated flies detected after extraction with LB1, LB2, or LB3 for both αSynWT and αSynA53T (Figure 6a,c and Appendix A). This is in agreement with the increased detection of αSyn observed by immunofluorescence after PQ treatment (Figure 5). Strikingly, the fraction of αSyn (αSynWT and αSynA53T) detected with the most stringent buffer LB3 strongly increased relative to LB1, which presumably originates from the global increase of αSyn levels and decreased detection in LB1 buffer (Figure 6b,d). This indicates that PQ promotes the accumulation of soluble and misfolded αSyn.

Given the increased misfolding of αSyn in response to PQ treatment, we next evaluated the resistance of αSyn to proteinase-K digestion. In contrast to aging flies (Figure 2a), we did not observe any increase of αSyn resistance to proteinase-K digestion in response to PQ treatment in flies expressing αSynWT or αSynA53T (Figure 7a–d). This suggests that PQ treatment maintains or promotes αSyn in a state still sensitive to proteinase-K digestion.

Interestingly, using limited proteolysis by proteinase-K [55,56], αSynA53T was better detected than in the absence of proteinase-K enzyme in both untreated and PQ-treated flies. This was observed as a pic of detection of αSyn for extracts treated with 0.5 μg/mL proteinase-K (Figure 7c,d). This was interpreted by the fact that proteinase-K makes a more aggregated αSynA53T fraction accessible, which is thus less accessible to immunodetection (“release effect”). Finally, the release effect was notably weaker in the presence of PQ, suggesting that αSyn is less aggregated in PQ-treated compared to untreated flies.

Finally, to complete the analysis on the effect of PQ on αSyn pathological status, we evaluated the total and phospho-synuclein levels (αSynp129) in paraffin section of flies expressing αSynWT or αSynA53T using the chronic exposure protocol of PQ at LT50. As in 60-day-old flies (Figure 1), we observed a global increase of αSyn level detected using the anti-αSyn antibody MJFR1 (Figure 8a,b,e,f). The levels of Ser129-phosphorylated αSyn were weak and broadly remained unchanged for αSynWT after PQ exposure (Figure 8c,d,i). In contrast, we observed a high immunoreactivity of αSynp129 (with EP1536Y antibody) for αSynA53T-expressing flies which was enhanced in the presence of PQ (Figure 8g,h,j). This accumulation of αSynp129 was particularly pronounced in the suboesophageal ganglion region (Figure 8h). Thus, PQ treatment induces significant accumulation of αSynp129 in flies expressing αSynA53T but not αSynWT. Collectively, these results demonstrate that PQ treatment promotes pathogenic modifications of αSyn in flies expressing αSynWT and αSynA53T. This includes an increased soluble and misfolded αSyn, which is not associated with an increased resistance to proteinase-K for both αSynWT and αSynA53T, as well as an increased αSynp129 detection for αSynA53T.

## 3. Discussion

In this report, we studied the effect of PQ on αSyn pathogenic modifications using a chronic model of PQ exposure in *Drosophila* expressing human SNCA gene in neurons. We first found that a mild dose of PQ (0.75 mM) exhibited a moderate fly toxicity (LT50 between 15 and 19 days) compared to acute treatment (10 or 20 mM depending on the studies), in which most of the flies die within 2–3 days [37,57]. Furthermore, PQ synergistically enhanced toxicity in αSynWT and αSynA53T *Drosophila*, compared to the control flies. Thus, the chronic exposure of PQ provides an appropriate model to study a synergistic toxic effect between PQ and αSyn in flies.

To study the effect of PQ on αSyn-expressing flies, we used a range of in situ and biochemical methods to detect and evaluate the α progressive conversion of Syn toward aggregated forms (Table 1). We first observed that PQ induced the accumulation of both αSynWT and αSynA53T proteins, which was associated with an increased detection of misfolded soluble αSyn using strong denaturating buffer. This result is in agreement with previous studies showing that PQ increased the level of αSyn in mice [33], in human neuroblastoma SH-SY5Y or in melanoma SK-MEL-2 cell lines [53]. Interestingly, PQ did not increase proteinase-K-resistant αSynWT or αSynA53T, suggesting that soluble misfolded αSyn accumulate but not fully aggregated forms in this paradigm. Particularly, we could not detect the accumulation of αSyn aggregates in situ in *Drosophila* as observed in mice expressing human SNCA and exposed to PQ [58]. In that study, we would like to emphasize, however, that the increase in proteinase-K-resistant αSyn induced by PQ have been reported in mice over-expressing αSyn after one month of PQ treatment. Therefore, the lack of detection of αSyn forms that are resistant to proteinase-K in flies exposed to PQ, may very well be due to an early step in αSyn aggregation that was not observed in the mice model. In support of our data, a study using a proximity ligation assay method, reported the detection αSyn oligomers that remain sensitive to proteinase-K digestion in human post-mortem brain sections [59]. It was suggested that oligomers accumulation, that are different from both physiological and highly aggregated αSyn, represent a very early event of αSyn pathology [59]. Indeed, high levels of αSyn soluble oligomeric forms, rather than the insoluble fibrillar forms, have been proposed to be the most pathogenic species in PD [60,61]. Especially, soluble αSyn oligomers can disrupt membranes [62,63] and cause cell death both in vitro [64,65] and in animal models [66,67]. Thus, we propose that PQ induces the toxic accumulation of αSyn, increases its insolubility but do not promote its aggregation toward highly aggregated forms in *Drosophila*.

PQ exposure was also associated with a pronounced accumulation of αSynp129 detected in the subesophageal ganglion region of the adult *Drosophila* brain in flies expressing αSynA53T. A specific increase of αSynA53T phosphorylation induced by PQ has been previously observed in transgenic mice, at least in the gut and after oral exposure [68]. However, we cannot exclude that the specific increase of αSynA53T phosphorylation induced by PQ is in part due to the genetic background of the *Drosophila* transgenic line, i.e., the co-expression of CG7900, a lipid droplet binding protein [69]. Further investigation will be required to determine the contribution of CG7900 to the pathogenic modifications of αSyn in PQ-treated flies.

Another interesting aspect raised in our study is whether PQ induces similar or different pathogenic modifications of αSyn than the ones observed in aging flies (Table 2). We first found that αSyn undergoes major modifications in aged compared to young flies, including: (1) the masking of αSyn epitopes detected by immunofluorescence in situ and by a weakly denaturating buffer in immunoblots, (2) an increased detection of αSyn by immunochemistry with an unmasking protocol or highly denaturating buffer by immunoblot, (3) an increased resistance to proteinase-K digestion and (4) a modest increase of αSyn phosphorylation. Our results show that αSyn modifications are quantitatively and qualitatively different during aging compared to a chronic exposure to PQ. The main differences are that aging but not PQ exposure promotes αSyn resistance to proteinase-K digestion, the masking of αSyn epitopes and a reduced detection in mild denaturating buffer. Furthermore, both PQ exposure and fly aging results in the accumulation of αSyn, but in greater extent with PQ exposure. Finally, the increased phosphorylation of αSyn observed during aging involves both αSynWT and αSynA53T was very modest compared to the specific increase in phosphorylation of αSynA53T in response to PQ. Collectively, these methods support that αSyn is converted to more aggregated forms during aging than upon PQ exposure. The fact that PQ promotes phosphorylated αSyn forms that do not exhibit proteinase-K resistance may correspond to more toxic forms, particularly observed with αSynA53T, which could explain the higher toxicity of PQ in the αSynA53T line. 

Collectively our results provide evidence that *Drosophila* provide a useful pre-clinical PD model to study the pathogenic modifications of αSyn during aging or under exposure to PQ or other environmental neurotoxins.

## 4. Materials and Methods

### 4.1. Drosophila Stocks and Harvesting

The following stocks were obtained from BDSC: *UAS-**αSynWT* (BL8146), *UAS-**αSynA53T* (BL8148), and the pan neuronal *ELAV-Gal4* driver (BL8760). The *UAS-GFP* was provided by Claude Desplan laboratory. Females from the ELAV-Gal4 line were crossed to males from the *UAS-**αSynWT*, *UAS-**αSynA53T* or *UAS-GFP*. Flies from each genotype were separated according to sex within 24–48 h post eclosion. For survival assays (Figure 1, Figure 2 and Figure 3), flies were raised on standard corn meal food supplemented with yeast, maintained at 25 °C, 60–65% humidity and a 12 h light/12 h dark cycle in a climate-controlled incubator. Male flies were used in the present study to avoid gender variance. Flies were transferred to new vials (~40 flies per vial) every three days. Flies were anesthetized with CO_2_ gas, transferred to tubes, then frozen at −80 °C until future use or deposited on dry ice to separate heads from the bodies.

### 4.2. PQ Treatment

Adult male flies, 48–96 h after eclosion, were fed on 10% sucrose and 0.8% agarose (3 mL) with or without PQ (0.75 mM). Fly viability assays were conducted at 25 °C. Fresh medium (10% sucrose and 0.8% agarose) was poured into vials 48 h before use and PQ (+PQ) or water (untreated) were added (100 µL) 24 h before adding the flies.

Groups of 25 age-matched flies were flipped to fresh vials containing PQ every 2–3 days. Fly vials were stored horizontally to avoid trapping flies in any liquids that may accumulate in the bottom of the vial. The number of dead flies was counted every day. Fly flipping to fresh vials containing PQ or control medium continued until all flies died or until the lethal time 50 (LT 50) for analysis. Kaplan-Meier lifespan curves were generated using GraphPad Prism v.6.07 (GraphPad Software, San Diego, CA, USA).

### 4.3. Immunostaining and Confocal Imaging

Flies were sedated on ice, decapitated before proceeding to brain dissection in a drop of HL3 medium [70] supplemented with D-Glucose (120 mM). Whole-mount dissected brains were fixed in paraformaldehyde (PFA) 4% and incubated in PBS supplemented with 0.5% Triton X-100 and 5 mg/mL bovine serum albumin (BSA). Primary antibodies including: rabbit anti-αSyn (C20R, ref sc-7011-R, Santa Cruz Biotechnology, Santa Cruz, CA, USA) (1:400), mouse anti αSyn-(Syn211, ref sc-12767, Santa Cruz Biotechnology, Santa Cruz, CA, USA) (1:400), and rabbit anti-αSyn (MJFR1, ref ab138501, Abcam, Cambridge, UK) (1:400) were diluted in the blocking solution overnight at 4 °C. The samples were washed 3 times with PBS-T and incubated overnight at 4 °C with anti-mouse Alexa Fluor 647 (1:400, A31571, Invitrogen, Thermo Fisher Scientific Inc., Waltham, MA, USA) or antirabbit Alexa Fluor 488 (1:400, A11008, Invitrogen, Thermo Fisher Scientific Inc., Waltham, MA, USA) or anti-mouse Alexa Fluor 488 (1:400, A32723, Invitrogen, Thermo Fisher Scientific Inc., Waltham, MA, USA) secondary antibodies diluted in blocking solution. Samples were washed three times and then mounted in Vectashield mounting medium (AbCys) on a bridge slide to prevent tissue flattening. Samples were stored at−20 °C until confocal microscopy.

### 4.4. Epitope Unmasking by Immunohistochemistry

Flies were decapitated on dry ice then heads were fixed in PFA 4% for 24 h and routinely processed in paraffin to produce histological sections (4 µm) for immunohistochemistry staining. The histological sections were deparaffinized and hydrated. To enhance the immunoreactivity of αSyn protein (epitope unmasking), sections were boiled in 0.1 mol/L citrate buffer (pH 5, 8) (ref C9999, Sigma-Aldrich, St. Louis, MO, USA) for 5 min in a microwave oven. The endogenous peroxidase activity was blocked by a 5-min immersion in a 3% hydrogen peroxide solution (ref 23615.261, VWR, Radnor, Wayne, PA, USA). The non-specific binding was blocked by incubating the sections in a solution of Blocking Reagent (ref 11096176001, Sigma-Aldrich, St. Louis, MO, USA) for 30 min. Subsequently, the sections were incubated overnight at 4 °C with primary antibodies rabbit anti-αSyn (MJFR1, ref ab138501, Abcam, Cambridge, UK) (1:500) or rabbit anti-αSynp129 (clone EP1536Y, ref ab51253, Abcam, Cambridge, UK) (1:300). The histological sections were blocked a second time by incubating the sections in the Solution Blocking Reagent (ref 11096176001, Sigma-Aldrich, St. Louis, MO, USA) for 30 min. The HRP-labeled anti-rabbit Ig secondary antibody (1:200) (ref 4010-05; CliniSciences, Nanterre, France) was used. HRP enzymatic reaction was revealed using the ImmPACT™ DAB substrate (ref SK-4105, CliniSciences, Nanterre, France). Sections were then counterstained with aqueous hematoxylin. The stained sections were observed under a light microscope BX51 (Olympus, Tokyo, Japan) coupled to INFINITY*3-6*UR camera (Lumenera, Ottawa, IL, Canada). Automated immunolabeling counting was performed with the Fiji software. Automated counting uses a threshold algorithm to distinguish and count staining from the background and determined the surface covered with stained.

### 4.5. Proteinase K Resistance Assay

Fly heads (n = 100) were homogenized on ice with a Dounce borosilicate glass grinder (0.1 mL) in a buffer containing (50 mM Tris–HCl, pH 7.5, 5 mM EDTA, NP40 1%, DTT 1 mM) and incubated 1 h at 25 °C. After centrifugation at 13.000 rpm for 1 min, supernatants were collected and incubated 30 min at 25 °C with indicated amounts of proteinase K (0, 0.5, 1, 1.5, and 2 µg/mL). Laemmli buffer was then added to the sample, before freezing.

### 4.6. Extraction of Soluble αSyn with Weak (LB1), Medium (LB2) and Strong Denaturating (LB3) Buffers

Fly heads (n = 100) were prepared in lysis buffers of increasing denaturating strength on ice with a Dounce borosilicate glass grinder (0.1 mL): LB1 (50 mM Tris–HCl, pH 7.5, 5 mM EDTA, NP40 0.5%, 1% phosphatase and protease inhibitor cocktails), LB2 (50 mM Tris–HCl, pH 7.5, 5 mM EDTA, NP40 1%, DTT 1 mM, 1% phosphatase and protease inhibitor cocktails) or LB3 (50 mM Tris–HCl, pH 7.5, 5 mM EDTA, NP40 0.5%, DTT 50 mM, 7M Urea, 2M thiourea, 1% phosphatase and protease inhibitor cocktails). After centrifugation at 13.000 rpm for 3 min, supernatants were collected. Laemmli buffer was then added to the sample, before freezing.

### 4.7. Western Blot Analysis

Proteins were separated by sodium dodecyl sulfate sulfate-polyacrylamide gel electrophoresis (SDS–PAGE: Stain-Free gel 4–15%, ref 5678084, Bio-Rad, Hercules, California, US) and transferred to polyvinylidene fluoride membranes. Membranes were then incubated with TBS containing 4% PFA and 0,01% glutaraldehyde for 30 min at room temperature before blocking with 3% BSA in TBS containing 0.1% Tween 20. The following antibodies were used: rabbit anti-αSyn (MJFR1, ref ab138501, Abcam, Cambridge, UK) (1:1000) and anti-β-tubulin (ref E7-c, Developmental Hybridoma Bank, Iowa City, IA, USA) (1:2000). After washing the membranes with TBS containing Tween 20, a horseradish peroxidase conjugated goat anti-mouse (ref 32460; ThermoFisher Scientific, Waltham, MA, USA) (1:1000) or anti-rabbit (ref 32430; ThermoFisher Scientific, Waltham, MA, USA) (1:1000) Ig secondary antibodies was applied. The immunocomplexes were visualized with chemiluminescent reagents (Supersignal WestDura, ref 34076, ThermoFisher Scientific, Waltham, MA, USA), followed by exposure on Biomax MR Kodak films, or CL-Exposure films, and by analysis using the Versa Doc system and Quantity One software (both from BioRad, Hercules, CA, USA).

### 4.8. Confocal Imaging and Image Processing/Analysis

Images of whole-mount brains were acquired using a Zeiss LSM800 confocal microscope at the LYMIC-PLATIM Imaging and Microscopy Core Facility of SFR Biosciences (UMS3444, ENS de Lyon, France). The lasers for αSyn detection were set up on aSynA53T expressing flies chronically exposed to PQ at the beginning of this study and kept unchanged for the quantification of both aging and exposure experiments.

For αSyn signal quantification, a projection of all of the Z-sections for the desired area (whole brain, antennal lobes or neuropils of the anterior ventrolateral protocerebrum) was made to sum the pixels intensity. For antennal lobes and neuropils of the anterior ventrolateral protocerebrum, we used the same Z thickness to generate the stack projection. The levels of fluorescence were determined on selected region of interest with Fiji. The fluorescence threshold was determined on a region that is outside of the labelled tissue. Intensity levels are expressed per unit area.

### 4.9. Statistical Analysis

Analyses of all data were conducted using GraphPad Prism v.6.07 (GraphPad Software, San Diego, CA, USA), using two-tailed Student’s t test assuming equal variances, using the Wilcoxon–Mann–Whitney two-sample rank-sum test or using a one-way ANOVA followed by the Dunnett’s post-test. Details of the analyses are described in the figure legends.

## Figures and Tables

**Figure 1 ijms-22-11613-f001:**
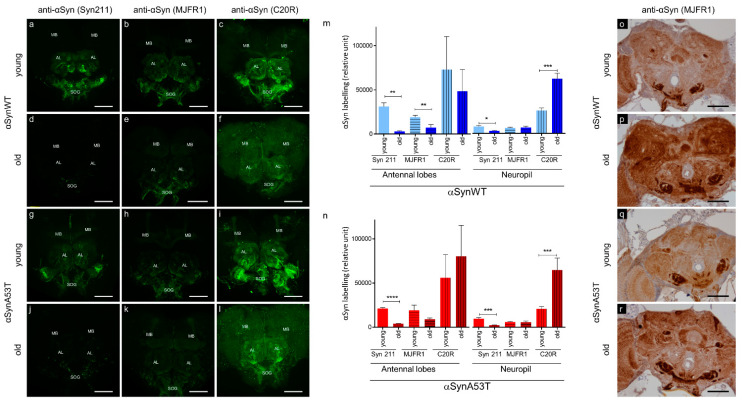
αSyn transgenic flies show an age-dependent αSyn epitope-masking. Representative immunofluorescence or immunohistochemistry brain images of young (5-day-old for immunofluorescence or 10-day-old for immunohistochemistry) (**a**–**c**,**g**–**i**,**o**,**q**) or old (60-day-old) (**d**–**f**,**j**–**l**,**p**,**r**) flies expressing human αsynWT (**a**–**f**,**o**,**p**) or αsynA53T (**g**–**l**,**q**,**r**) under the ELAV driver. Flies were raised on standard corn meal food supplemented with yeast. For immunofluorescence images, αSyn was stained (indicated in green) with different αSyn-specific antibodies: Syn 211 (**a**,**d**,**g**,**j**), MJFR1 (**b**,**e**,**h**,**k**) or C20R (**c**,**f**,**i**,**l**) on dissected fly brains of young and old flies. Images were projected from confocal Z-stacks acquired on confocal microscope using MAX intensity method. Note a prominent staining of the αSyn in the mushroom bodies (MB), cell bodies of the antennal lobes (AL) and suboesophageal ganglions (SOG). Quantifications of fluorescence were determined after z-projection (using SUM intensity method) on selected regions of interest (antennal lobes or in neuropils of the anterior ventrolateral protocerebrum) for flies expressing human αsynWT (**m**) or αsynA53T (**n**). Data are shown as mean with standard deviation for the three different αSyn-specific antibodies for young or old flies from two independent experiments: Syn211 (n = 5 for αsynWT and n = 6 for αsynA53T in antennal lobes; n = 5 for αsynWT and n = 6 for αsynA53T in neuropil), MJFR1 (n = 6 for αsynWT and n = 6 for αsynA53T in antennal lobes; n = 7 for αsynWT and n = 6 for αsynA53T in neuropil) and C20R (n = 7 for αsynWT and n = 9 for αsynA53T in antennal lobes; n = 7 for αsynWT and n = 8 for αsynA53T in neuropil). *p*-values of the group differences were calculated using *t*-test (** *p* = 0.0026 (αsynWT, antennal lobes, Syn211), ** *p* = 0.0070 (αsynWT, antennal lobes, MJFR1), **** *p* < 0.0001 (αsynA53T, antennal lobes, Syn211), * *p* = 0.0138 (αsynWT, neuropil, Syn211), *** *p* = 0.0002 (αsynWT, neuropil, C20R), *** *p* = 0.0009 (αsynA53T, neuropil, Syn211) and *** *p* = 0.0006 (αsynA53T, neuropil, C20R), respectively). Immunohistochemistry was performed using MJFR1 antibody (**o**–**r**) with antigenic unmasking protocol and revealed an accumulation of αSyn in older flies. Scale bar, 30 µm.

**Figure 2 ijms-22-11613-f002:**
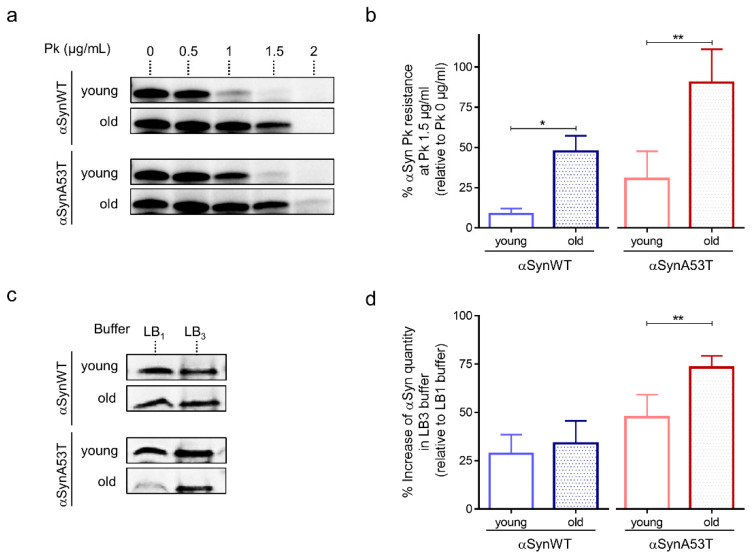
Age-dependent biochemical profile of αSyn protein in αSyn-expressing flies. (**a**) Proteinase-K (PK) resistance assay of αSyn performed on protein extracts from heads of αSynWT and αSynA53T expressing flies analyzed by Western blot with the anti-αSyn MJFR1 antibody. The lanes show increasing concentration of PK (from 0 to 2 µg/mL) for young (1-day-old) or old (40-day-old) flies. (**b**) The graph shows the percentage of protein remaining after digestion with 1.5 µg/mL PK concentration (relative to the undigested sample) in young (before 20-day-old) or old (at least 40-day-old) for αSynWT and αSynA53T expressing flies. Western blot signals were measured by densitometry using the software Image lab version 5.2.1 build 11. Quantifications showed a significant increase in αSyn PK resistance between young flies and old flies for αSynWT (n = 4) and αSynA53T (n = 7). Data are shown as means ± standard deviations. *p*-values of the group differences were calculated using *t*-test (* *p* = 0.0335 and ** *p* = 0.0086, respectively). (**c**) Soluble αSyn protein detection after extraction with denaturating buffers of increasing strength (LB1 < LB3). Western blot assessment (with the MJFR1 antibody) of αSyn protein in the soluble fraction of LB_1_ 0.5% NP40 and LB_3_ urea/thiourea lysis buffers on young (1-day-old) or old (40-day-old) flies expressing αSynWT or αSynA53T. (**d**) The graph shows the percentage of protein quantity in LB3 relative to LB1 for young and old flies (n = 7 for αSynWT and n = 8 for αSynA53T). Data are shown as mean with standard deviation. *p*-values of the group differences were calculated using *t*-test (** *p* = 0.005). Flies were raised on standard corn meal food supplemented with yeast.

**Figure 3 ijms-22-11613-f003:**
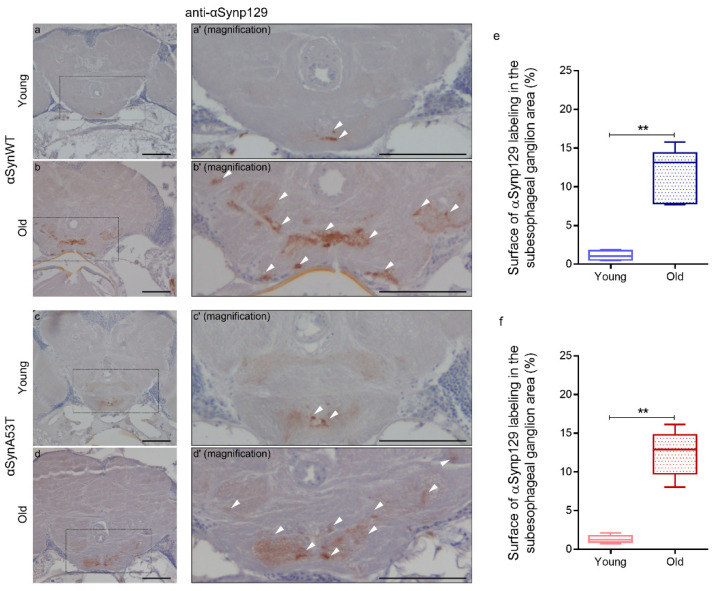
Age-dependent increase of αSynp129 in αSyn-expressing flies. Representative immunohistochemistry paraffin-embedded brains stained with an anti-αSynp129 (EP1536Y) antibody on young (5 day-old) (**a**,**c**) or old (60 day-old) (**b**,**d**) flies expressing αSynWT (**a**,**b**) or αSynA53T (**c**,**d**). Magnification of area indicated in (**a**–**d**) are shown in (**a’**–**d′**). Note that localized staining of the αSynp129 is observed in the suboesophageal ganglion area as well as an increased detectability in older flies. White arrowheads indicate αSynp129 deposits. Scale bar represents 30 μm. Quantifications show the surface of αSynp129 labeling (EP1536Y antibody) in subesophageal ganglion area for αSynWT (**e**) or αSynA53T (**f**) in young (5- or 10-day-old) or old (50- or 60-day-old) flies. In the box and whisker plots in (**e**,**f**), boxes extend from the first to the third quartile, the line inside the boxes shows the median and the whiskers represent the min/max value of independent experiments: αSynWT (n = 4 for young, n = 6 for old) or αSynA53T (n = 5 for young, n = 6 for old). *p*-values of the group differences were calculated using Mann–Whitney test (** *p* = 0.0095 for αSynWT and ** *p* = 0.0043 for αSynA53T). Flies were raised on standard corn meal food supplemented with yeast.

**Figure 4 ijms-22-11613-f004:**
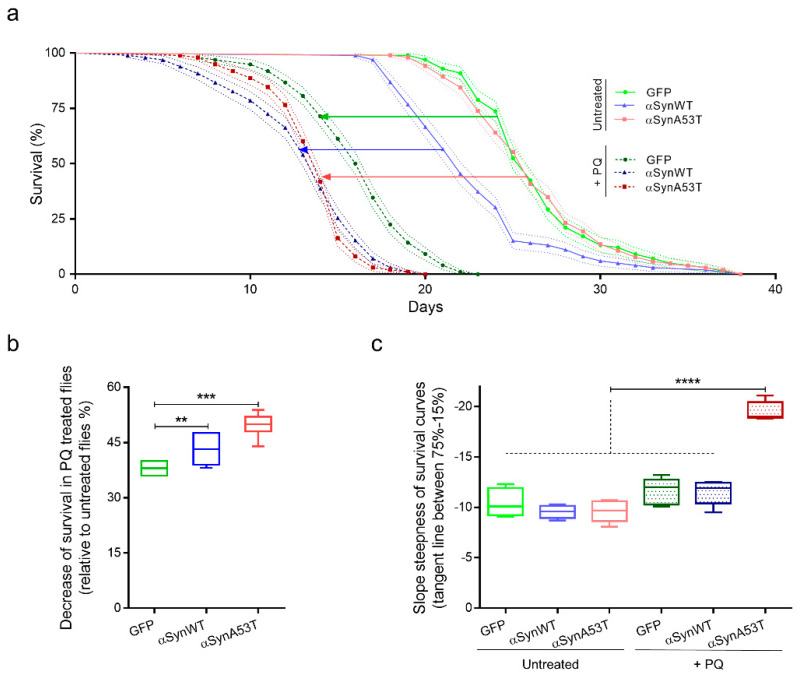
Chronic exposure to PQ induces a synergic toxicity with αSyn on transgenic *Drosophila*. (**a**) Survival curves of *Drosophila* males raised in control medium (agarose + 10% sucrose diet), expressing GFP (light green), αsynWT (light blue), αsynA53T (light red) or raised in control medium supplemented with PQ (0.75 mM), expressing GFP (darker green), αsynWT (darker blue), and αsynA53T (darker red). Standard errors are represented for each survival curve as dashed light lines, n = 5 independent experiments for a total of 6861 flies. (**b**) Decrease of fly survival after PQ exposure. Statistical significance of differences (Mann–Whitney U-test) compared with GFP group, *** *p* = 0.0002 and ** *p* = 0.0087 respectively, n = 8. (**c**) Slope steepness of survival curves calculated between 75% and 15%. Statistical analysis was performed using Dunnett’s test for multiple comparisons to evaluate intergroup differences, **** *p* < 0.0001, n = 5. In the box and whisker plots in (**b**,**c**), boxes extend from the first to the third quartile, the line inside the boxes shows the median and the whiskers represent the min/max value of independent experiments.

**Figure 5 ijms-22-11613-f005:**
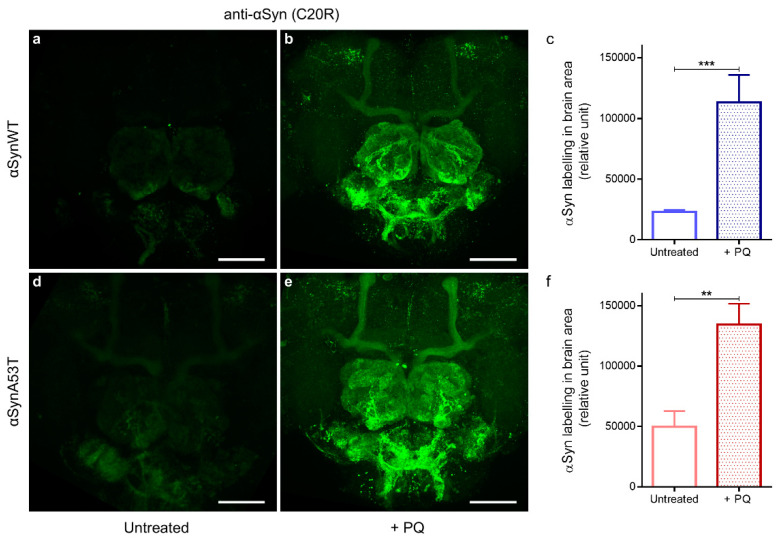
Chronic exposure to PQ causes αSyn accumulation in αSyn expressing flies. Representative immunofluorescence images of dissected brains stained with an anti-αSyn (C20R) antibody of αSynWT and αSynA53T expressing flies raised in control (**a**,**d**) or PQ supplemented medium (**b**,**e**) and analyzed at LT50. Images were projected from confocal Z-stacks acquired on confocal microscope. Scale bar, 30 µm. Quantifications showed a significant increase in αSyn detection between flies exposed to PQ and untreated flies for αSynWT (n = 6) (**c**) and αSynA53T (n = 9) (**f**). Data are shown as mean with standard deviation. *p*-values of the group differences were calculated using *t*-test (*** *p* = 0.0003 and ** *p* = 0.0059, respectively).

**Figure 6 ijms-22-11613-f006:**
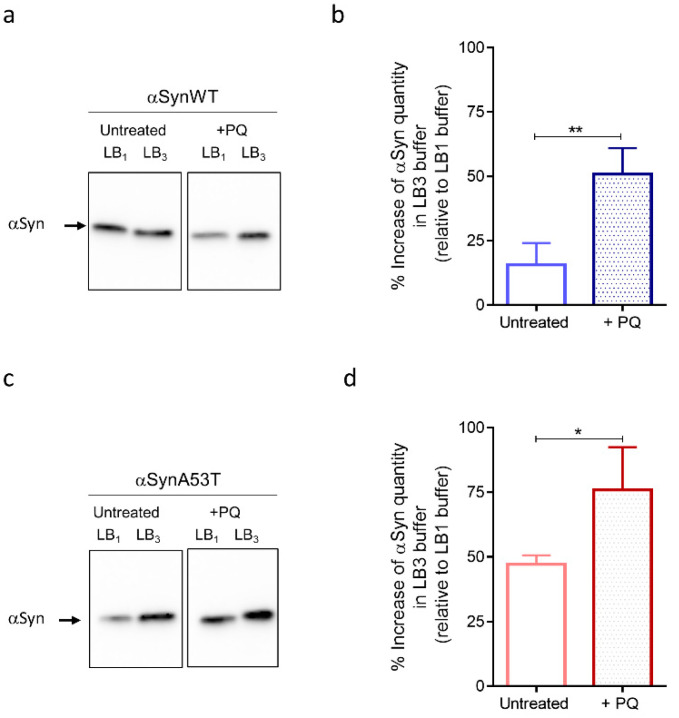
Fly chronic exposure to PQ affects soluble αSyn detection in denaturating buffers. Western blot (antibody MJFR1) analysis on fly head extracts from control (untreated) or PQ exposed (+PQ) flies expressing αSynWT (**a**,**b**) or αSynA53T (**c**,**d**) realized at LT50. Heads (n = 100) were homogenized using a lysis buffer 1 (LB10.5% NP40) or a chaotropic lysis buffer 3 (LB3 urea/thiourea). The optical density of each sample was measured and normalized using a β-tubulin run on the same gel. The graphs are composites of six independent experiments (n = 3 for αSynWT and n = 3 for αSynA53T). The data, shown as mean with standard deviation in (**b**,**d**), represent the increase of the signal quantify with LB3 urea/thiourea compared to LB10.5% NP40 for a same experimental condition. *p*-values of the group differences were calculated using *t*-test (** *p* = 0.0045 for αSynWT and * *p* = 0.0203 for αSynA53T).

**Figure 7 ijms-22-11613-f007:**
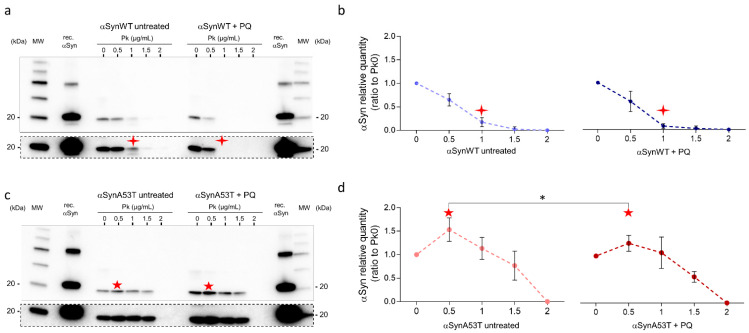
Fly chronic exposure to PQ promotes αSyn proteinase-K-sensitive forms. Heads (n = 100) from αSynWT (**a**,**b**) and αSynA53T expressing flies (**c**,**d**) at LT50 were digested with proteinase-K (PK) (25 °C, 30 min) at increasing concentrations and analyzed by Western blot with an anti-αSyn (MJFR1) antibody. Cropped images surrounded by dashed lines from the corresponding condition gel with a longer exposition are shown below each gel. Densitometric analyses (**b**,**d**) are indicative of 10 independent experiments corresponding to: untreated αSynWT (n = 4), PQ αSynWT (n = 4), untreated αSynA53T (n = 6), and PQ αSynA53T (n = 7). Red stars show the modifications of the PK digestion profile observed in αSynWT (four-pointed stars) or αSynA53T (five-pointed stars) expressing flies. *p*-values of the group differences were calculated using *t*-test (* *p* = 0.0470).

**Figure 8 ijms-22-11613-f008:**
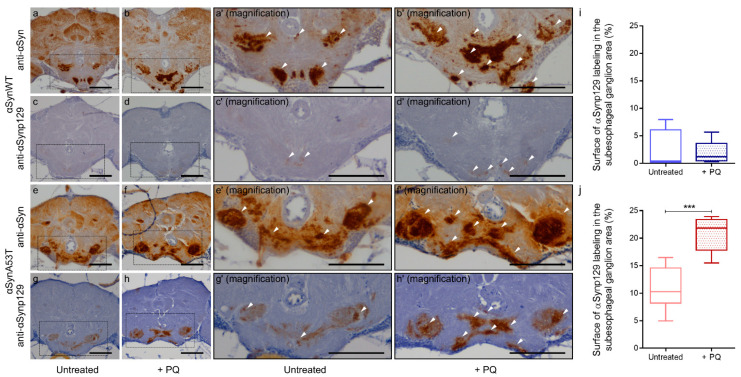
Chronic exposure to Paraquat induces the accumulation of αSynp129. In brains of αSynA53T-expressing flies. Immunohistochemistry was performed using anti-αSyn (MJFR1) (**a**,**b**,**e**,**f**) or anti-αSynp129 (EP1536Y) (**c**,**d**,**g**,**h**) antibodies on paraffin-embedded brain sections of αSynWT (**a**–**d**) or αSynA53T (**e**–**h**) expressing flies untreated or exposed to PQ (+PQ) at LT50. Magnification of area indicated in (**a**–**d**) are shown in (**a’**–**d′**). Magnification of area indicated in (**e**–**h**) are shown in (**e’**–**h′**). White arrowheads indicate αSyn or αSynp129 deposits. The scale bar is 30 μm. Quantifications show the surface of αSynp129 (EP1536Y) labeling in subesophageal ganglion area for αSynWT (**i**) or αSynA53T (**j**) in control (untreated) or PQ exposed (+PQ) flies. In the box and whisker plots in (**i**,**j**), boxes extend from the first to the third quartile, the line inside the boxes shows the median and the whiskers represent the min/max value of 10 independent experiments: αSynWT (n = 4 for untreated, n = 5 for PQ) or αSynA53T (n = 8 for untreated, n = 10 for PQ). *p*-values of the group differences were calculated using the Mann–Whitney test (*** *p* = 0.0002).

**Table 1 ijms-22-11613-t001:** In situ and biochemical assays to follow αSyn pathogenic modifications.

	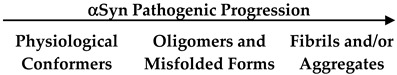	
αSyn Modifications	References
Serine 129 phosphorylation	Low	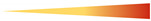	High	[4,39]
Resistance to mild denaturating buffer	None	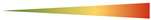	High	[40,41,42,43]
Structural rearrangements	None	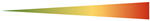	High	[44,45]
Resistance to PK digestion	None	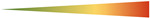	High	[46,47,48]

**Table 2 ijms-22-11613-t002:** Comparison of αSyn pathogenic progression induced during normal aging or by PQ exposure. αSyn pathogenic modifications in aged (40- to 60-day-old) or young PQ exposed flies (LT50 between 15- and 19-day-old) in flies expressing human αSyn (αSynWT or αSynA53T) under the ELAV driver.

Modifications (αSyn)	Aging-Induced	PQ-Induced
Accumulation	Yes (αSynWT, αSynA53T)	Yes (αSynWT, αSynA53T)
Serine 129 phosphorylation	Yes (αSynWT, αSynA53T)	Undetected or Yes (αSynWT) (αSynA53T)
Resistance to mild denaturating buffer	Undetected or Yes (αSynWT) (αSynA53T)	Yes (αSynWT, αSynA53T)
Masking of αSyn epitopes	Yes (αSynWT, αSynA53T)	Undetected (αSynWT, αSynA53T)
Resistance to PK digestion	Yes (αSynWT, αSynA53T)	Undetected (αSynWT, αSynA53T)

## Data Availability

Data is contained within the article or Appendix A.

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
