# Peer review of "Chronic Exposure to Paraquat Induces Alpha-Synuclein Pathogenic Modifications in Drosophila"

_ijms, 2021, doi:10.3390/ijms222111613_

Round 1

Reviewer 1 Report

The Arsac et al. investigated an interesting two factors Drosophila model for Parkinson's Disease (PD). They have combined chronic exposure of low-dose of paraquat with the human a-Syn transgenetic fly model to probe the pathological modifications of a-Syn (WT and A53T). In addition, the manuscript compared the paraquat treatment with fly aging with multiple biochemical methods.  

The authors have reported a significant model that involves both genetic predisposition and environmental toxicity. They provided a set of complementary methods to evaluate the biochemical profile of a-Syn protein in the flies. However, the interpretation of some results was unclear, and the paper lacked repetition and quantification in Figures 1-3 for fly aging analysis. 

Major concerns:

  1. In Figure 1, the authors want to demonstrate the age-dependent a-Syn epitope-masking with only schematic representations. Repetition and quantification are needed to convince the readers that both a-Syn WT and A53T were misfolded during aging in this model. Also, it would be interesting to statistically compare the unmasked a-Syn WT to the a-Syn A53T levels in the neuropile by immunochemistry. 
  2. Similar concerns in Figures 2 and 3, the authors need to support the conclusions with repetition and quantification.  
  3. In Figures 5 C and F, the quantification method of the a-Syn labeling in the brain area was unclear. The authors should specify the parameters such as the fluorescence threshold in image analysis for unbiased comparison in Methods.
  4. In lines 298-299, the authors need to specify the reasoning behind the conclusion that paraquat promotes the accumulation of soluble and misfolded a-Syn indicated by the higher percentage increase of a-Syn quantity in LB3 relative to in LB1. Given that the total a-Syn quantity increased with paraquat treatment, the comparison between internally normalized increases was not logically conclusive (the % increase could result from a decrease in LB1 as in Figure 2B, or a significant increase in LB3 indicated in Figure 6C).
  5. In lines 380-381, the authors equalized the proteinase-K resistant a-Syn to a-Syn fibrils or aggregates(?). However, the link between the a-Syn morphology and solubility is ill-defined in the in-vivo models. It was reported that fibrils were proteinase-K resistant. Nevertheless, no clear evidence indicated the morphology of proteinase-K resistant oligomers must be fibrils. 

Minor concerns:

  1. In line 14, the toxic mechanism induced by paraquat (in PD) has been extensively investigated (see Tanner et al., 2011 in EHP). The authors might need to change the wording to emphasize the ill-defined relationship between paraquat and a-Syn in PD pathology. 
  2. In line 73, it should be UAS/GAL4; the "4" was missing.
  3. In Figure 2, the authors stated that the old flies used in the Western blot were 40 days old. However, the survival curves in Figure 4 indicated a 0% survival rate in all groups before day 40. 
  4. In Figure 4, the survival curves indicate that untreated a-Syn WT flies died earlier than a-Syn A53T and GFP control. The authors need to explain or suggest the possible underlying mechanisms.
  5. In Figures 5 C and F, the authors need to discuss that the results indicated that the a-Syn A53T expression produced significantly more a-Syn in the flies' brains when comparing two untreated groups. 
  6. In line 324, the authors need references for the "release effect" to explain the pic of detection of a-Syn in proteinase-K resistance assay in Figure 7D.

Author Response

Reviewer #1

The Arsac et al. investigated an interesting two factors Drosophila model for Parkinson's Disease (PD). They have combined chronic exposure of low-dose of paraquat with the human a-Syn transgenetic fly model to probe the pathological modifications of a-Syn (WT and A53T). In addition, the manuscript compared the paraquat treatment with fly aging with multiple biochemical methods.  

The authors have reported a significant model that involves both genetic predisposition and environmental toxicity. They provided a set of complementary methods to evaluate the biochemical profile of a-Syn protein in the flies. However, the interpretation of some results was unclear, and the paper lacked repetition and quantification in Figures 1-3 for fly aging analysis. 

We would like to thank the reviewer for his comments, which helped us improving the manuscript. As suggested, we have included quantifications of the results presented in Figure 1-3. This novel presentation of the results strengthens our initial claims. In response to his comments, we have also provided more details to explain our method of quantification of aSyn fluorescent labeling in the Material and Methods. Finally, we have addressed all the concerns in his remaining comments on the interpretation of our data. We hope that the reviewer will be satisfied by our answers.

Major concerns:

1. In Figure 1, the authors want to demonstrate the age-dependent a-Syn epitope-masking with only schematic representations. Repetition and quantification are needed to convince the readers that both a-Syn WT and A53T were misfolded during aging in this model. Also, it would be interesting to statistically compare the unmasked a-Syn WT to the a-Syn A53T levels in the neuropile by immunochemistry. 

We agree with the reviewer that current schematic representation of labeling detection in Figure 1 deserves quantifications. In response to the reviewer’s comment, we now provide quantification of immunofluorescent aSyn staining in young flies expressing aSynWT or aSynA53T compared with older flies of the same genotype with Syn 211, MJFR1 and C20R (Figure 1A-1L). The result of the quantifications of aSyn labeling in the antennal lobes and neuropile confirmed our initial observations but allowed us to bring a more precise description of the variation in the detection with the different antibodies. In particular, we report a more intense immunofluorescent  aSyn staining in young flies expressing aSynWT or aSynA53T compared with older flies of the same genotype with the Syn 211 antibody (Figure 1A, D, G, J, M and N). Interestingly, for the MJFR1 antibody, the loss of aSyn labeling in Drosophila brain during aging is less obvious in antennal lobes for aSynA53T and is not observed in neuropil in flies expressing aSynWT or aSynA53T (Figure 1M-N). In contrast to Syn 211 and MJFR1 antibodies, the labeling of aSynWT and aSynA53T with the C20R antibody was still visible in antennal lobe and was associated with a weak and uniform increase in the signal of aSyn labeling in the whole neuropil (Figure 1F, L M and N). This may be due to the fact that the C20R antibody readily detects aSyn forming aggregates, as shown by detection of both SDS-resistant oligomers in Western blot and specific ELISA immunoreactivity of such aggregates produced in vitro by Protein Misfolding Cyclic Amplification (Nicot et al. 2019, PMID  31370680). Collectively, these results demonstrate variability in the immunodetection of aSyn labeling in Drosophila brain during aging. This raises the possibility that aSyn epitopes detected by Syn 211 (111-125 amino acid sequence) and to a lesser extent MJFR1 (118-123 amino acid sequence) but not C20R (C-terminal unknown amino acid sequence) antibodies could be masked due to the change of aSyn conformation during aging. The result section has been updated in pages 7-8 to reflect those changes. We have also updated the legend of Figure 1 to detail the repetition used for the quantification of each genotype (page 23). 

We also agree with the reviewer that the quantification of aSyn detected by immunochemistry after the unmasking protocol, would be ideal. However, with the DAB staining given that the labeling spreads in most of brain region already in young flies, signal quantification is not adaptable/reliable? in this case. We thus cannot determine the surface of aSyn labeling. Thus, we propose a qualitative reading of IHC (and accordingly remove schematic representations results) associated with quantitative results for the neuropil by IF. The fact that aSynWT and aSynA53T are strongly detected in old flies using an unmasking protocol followed by IHC, suggests that aSyn is misfolded during aging. These results are also supported by aSyn increased resistance to proteinase K and its increased detection in stringent buffer (LB3) compared to mild detergent (LB1) (Figure 2).  In conclusion, we hope that the reviewer will be convinced by the modifications provided in Figure 1, which strengthen our message.  

2. Similar concerns in Figures 2 and 3, the authors need to support the conclusions with repetition and quantification.  

In response to the reviewer’s comments, we have included quantifications of the data presented in Figures 2 and 3.

In Figure 2, we have included a quantification of the proportion of aSyn resistant to proteinase-K digestion at 1.5mg/mL in young and old flies expressing aSynWT or aSynA53T (Figure 2B). We now show the same representative immunoblot with longer exposure time, which best illustrates the quantification. This new layout comforts our initial claim that aSyn becomes resistant to proteinase K digestion during fly aging. We have updated the legend of Figure 2 to detail the sample number used for the quantification in each condition.

We have also quantified the result of aSyn detection using extraction buffers of increasing denaturating strength. The proportion of aSynA53T quantity detected with LB3 relative to LB1, ranged from 47% to 73% in young and old flies, respectively (Figure 2D). The reduced detection in LB1 but higher detection in LB3 buffers suggests a progression of aSynA53T into the aggregative process in old flies. We have also updated the figure 2 legend to reflect the sample number used for the quantification in each condition. 

In Figure 3, we have included the quantification of surface area covered by pSyn129 labeling by immunochemistry in brains of flies expressing aSynWT and aSynA53T (Figure 3E and 3F). The quantification comforts our initial observation that aSynp129 levels increase in these two conditions in old compared to young flies (Figure 3B’, D’, E and F).

3. In Figures 5 C and F, the quantification method of the a-Syn labeling in the brain area was unclear. The authors should specify the parameters such as the fluorescence threshold in image analysis for unbiased comparison in Methods.

We recognize that our description was too brief. We propose the addition of a paragraph” to complete our methodological description, as followed in the section “Confocal imaging and image processing/analysis” of the Material and Methods : “Images of whole-mount brains were acquired using a Zeiss LSM800 confocal microscope at the LYMIC-PLATIM Imaging and Microscopy Core Facility of SFR Biosciences (UMS3444, ENS de Lyon, France). The lasers for αSyn detection were set up on aSynA53T expressing flies chronically exposed to PQ at the beginning of this study and kept unchanged for the quantification of both aging and exposure experiments.

For αSyn signal quantification, a projection of all of the z-sections for the desired area (whole brain, antennal lobes or neuropil) is made to sum the pixels intensity. For antennal lobes and neuropil, we used the same Z thickness to generate the stack projection. The levels of fluorescence were determined on selected region of interest with Fiji. The fluorescence threshold was determined on a region that is close to the defined region of interest without fluorophore specifically bound to a target. Intensity levels are expressed per unit area. “

4. In lines 298-299, the authors need to specify the reasoning behind the conclusion that paraquat promotes the accumulation of soluble and misfolded a-Syn indicated by the higher percentage increase of a-Syn quantity in LB3 relative to in LB1. Given that the total a-Syn quantity increased with paraquat treatment, the comparison between internally normalized increases was not logically conclusive (the % increase could result from a decrease in LB1 as in Figure 2B, or a significant increase in LB3 indicated in Figure 6C).

We agree that in principle, it would be possible that given the increase quantity of aSyn in response to PQ, the observed higher percentage increase of aSyn quantity in LB3 relative to in LB1 could result from a decrease in LB1 detection and/or an increase in LB3. However, based on our observations, a decreased detection in LB1 is always associated with an increased detection of aSyn with LB3, which supports the accumulation of soluble and misfolded aSyn.

To clarify our reasoning, we propose the following modified sentence in the result section (page 12):

“Strikingly, the fraction of aSyn (aSynWT and aSynA53T) detected with the most stringent buffer LB3 strongly increased relative to LB1, which presumably originates from the global increase of aSyn levels and decreased detection in LB1 buffer (Figure 6B and 6D). This indicates that PQ promotes the accumulation of soluble and misfolded aSyn.”

5. In lines 380-381, the authors equalized the proteinase-K resistant a-Syn to a-Syn fibrils or aggregates(?). However, the link between the a-Syn morphology and solubility is ill-defined in the in-vivo models. It was reported that fibrils were proteinase-K resistant. Nevertheless, no clear evidence indicated the morphology of proteinase-K resistant oligomers must be fibrils. 

We agree with the reviewer that the increased resistance to proteinase K does not provide information on the morphology of a-Syn. We propose to remove the term “fibrils” by “fully aggregated forms” in the following sentence of the discussion (page 14): “Interestingly, PQ did not increase proteinase-K-resistant aSynWT or aSynA53T, suggesting that soluble misfolded aSyn accumulate but not fully aggregated forms in this paradigm”.

Minor concerns:

  1. In line 14, the toxic mechanism induced by paraquat (in PD) has been extensively investigated (see Tanner et al., 2011 in EHP). The authors might need to change the wording to emphasize the ill-defined relationship between paraquat and a-Syn in PD pathology. 

We agree with the reviewer that PQ is a known pesticide that induce oxidative stress in exposed individuals (Tanner et al. 2011). We propose to modify the sentence of the abstract as followed: “PQ is known to cause an increased oxidative stress in exposed individuals but the consequences of such stress on aSyn conformation remains poorly understood.”

2. In line 73, it should be UAS/GAL4; the "4" was missing.

Thank you, we have corrected that mistake.

3. In Figure 2, the authors stated that the old flies used in the Western blot were 40 days old. However, the survival curves in Figure 4 indicated a 0% survival rate in all groups before day 40. 

We would like to thank the reviewer for noticing this error and we apologize for the misunderstanding. This is probably due to a typo mistake in the “Material and Methods”, in which we indicated that the 40 days old flies used for the Western blot came from Figure 4 analysis instead of from the analyses realized in Figures 1, 2 and 3. Indeed, as pointed by the reviewer, the flies used in figure 4 are all dead before day 40, which is due to the fact that flies were fed flies were fed on 10% sucrose and 0.8% agarose diet. This is in contrast to the experiments realized for Figure 1-3, in which flies were raised on standard corn meal food supplemented with yeast, which explains the longer lifespan in this condition compared to the agarose/sucrose diet.  We have corrected this mistake in the Material and Methods.  In addition, we propose to clarify the use of standard fly food in the legends of Figures 1, 2 and 3.

4. In Figure 4, the survival curves indicate that untreated a-Syn WT flies died earlier than a-Syn A53T and GFP control. The authors need to explain or suggest the possible underlying mechanisms.

We also noticed that aSynWT flies died earlier than aSynA53T or GFP control. This was somehow expected that aSynWT exhibit more toxicity than GFP expressing flies. However, it was also surprising to us that aSynA53T did not show intrinsic toxicity compared to GFP, while in presence of PQ, aSynA53T conferred an enhanced toxicity. We do not have clear explanation to explain that untreated flies expressing aSynA53T are as healthy as control but this could due to the genetic background associated with the P(UAS-aSynA53T) insertion which diminishes the intrinsic toxicity of aSynA53T. We propose to add the following sentence in the result section (page 11): “The fact that aSynWT, but not aSynA53T, exhibits a higher mortality than control flies is not clear but we cannot exclude an effect due to the different genetic insertions of the P[UAS] transgenes”.

5. In Figures 5 C and F, the authors need to discuss that the results indicated that the a-Syn A53T expression produced significantly more a-Syn in the flies' brains when comparing two untreated groups. 

We agree with the reviewer that aSynA53T levels are a bit higher than aSynWT levels in untreated conditions. This could be due to an increased stability of aSynA53T compared to aSynWT. Alternatively, as discussed in the previous comment, we cannot exclude an effect due to the different genetic insertions of the P[UAS] transgenes as described also in the discussion (page 15). Considering that the difference in aSyn is small compared to the increased levels after PQ treatments, we propose not to insist here on this aspect which otherwise would break the flow in the text.

6. In line 324, the authors need references for the "release effect" to explain the pic of detection of a-Syn in proteinase-K resistance assay in Figure 7D.

We thank the reviewer for letting us clarify what we named the “release effect”. It corresponds to an increased detection of aSyn in the presence of mild dose of proteinase-K or limited proteolysis. This approach has been previously used in biochemical studies to determine domain organization, folding properties, and ligand binding activities of proteins. In addition, limited proteolysis can provide important information about folding intermediates (Fontana et al., 2004 PMID 15218531) and has been used to study molecular features of protein aggregates associated with severe diseases, such as Alzheimer's disease and Parkinson's disease, and type 2 diabetes (Polverino de Laureto et al., 2003 PMID 14596805).

We propose to update the text with these references as followed:

“Interestingly, using limited proteolysis by proteinase-K (Fontana et al. 2004, Polverino de Laureto et al. 2003), aSynA53T was better detected than in the absence of proteinase-K enzyme in both untreated and PQ-treated flies.”

Reviewer 2 Report

Review of a manuscript “Chronic exposure to paraquat induces alpha-synuclein pathogenic modifications in Drosophila” by Arsac and coauthors submitted to IJMS.

Parkinson’s disease is a severe neurodegenerative diseases associated with accumulation of neuronal intracellular aggregates composed of alpha-synuclein and other proteins. There is no efficient treatment affecting the course of the disorder and no reliable biomarkers for early diagnosis of this pathology. So basic study using various models of Parkinson’s disease are very important since they may identify new targets for the treatment and new diagnostic biomarkers. The authors investigated alpha-synuclein pathogenic modifications in response to paraquat in a Drosophila model of Parkinson’s disease. The authors found that the chronic exposure of paraquat in fly expressing human alpha-synuclein is an efficient approach to reveal new mechanisms of environmental neurotoxins action on Parkinson’s pathology. This field of investigation is very important and the results presented in the manuscript will be interesting for the readers of IJMS.

The following corrections should be made:

Abstract

Line 21: “..but that do not form fibrils or aggregates that are detected by the proteinase K assay”. The sense of this sentence is unclear, because something is missed here. Do the authors want to say: “..but that they do not form fibrils or aggregates” or “..but that these toxic forms of aSyn do not form fibrils or aggregates”. Should be clarified.

Introduction:

Line 28-29: “Syn is a vertebrate-specific 140 amino-acid presynaptic protein, which can acquire neurotoxic properties during aging and in synucleopathies such as PD.” The authors should correct “synucleopathies” on a more commonly used term “synucleinopathies” and cite here a review: ”Intracellular dynamics of synucleins: Here, there and everywhere. International Review of Cell Molecular Biology, 2015; 320:103-169.”

Line 35: ”The accumulation of misfolded a-Syn…” the authors should add here a citation to a recent article:  “Cell responses to extracellular α-synuclein. Molecules 2019, 24 (2), 305, 21.

Line 46: ”With non-human primates that most closely mimic the human pathology, mouse is the most used animal model…” The sense of the sentence is unclear. How non-human primates and mouse are combined here? May be the authors mean: ” Although non-human primates most closely mimic the human pathology, mouse is the most often used animal model”.

Results:

Line 126: “the detection of aSyn aggregates is not routinely used due their difficult detection in tissues of Drosophila transgenic lines” should be corrected as follows:” the detection of aSyn aggregates is not routinely used due to difficulties in their detection in tissues of Drosophila transgenic lines”

Line 133: ”We observed a more intense aSyn staining by immuno-fluorescence” This is a clumsy sentence which may be written as follows :” We observed a more intense immunofluorescent aSyn staining…”

Line 186: ”This result suggests that the misfolding or aggregation of aSyn A53T in old flies is responsible for its reduced detection in LB1 buffer.” This is an interesting assumption, which requires more detailed explanation. What is in favor of misfolding and what is in favor of aggregation?

Line 232: ”…typically between 10 and 20mM fed to adult flies” The acute dose of PQ used is not clearly described here. If the authors used solution of PQ with concentration 10 and 20 mM, what amount of PQ the flies received?  Should be explained more precisely.  

Figure 6C. The arrow with a-Syn is located at higher position than protein band on this figure.

   Discussion

Line 367-368 – similar to comment to line 232: 0.75mM - a moderate fly toxicity, 10-20 mM acute treatment. Could the authors described more precisely what dose the flies received?

Author Response

Reviewer #2

Review of a manuscript “Chronic exposure to paraquat induces alpha-synuclein pathogenic modifications in Drosophila” by Arsac and coauthors submitted to IJMS.

Parkinson’s disease is a severe neurodegenerative diseases associated with accumulation of neuronal intracellular aggregates composed of alpha-synuclein and other proteins. There is no efficient treatment affecting the course of the disorder and no reliable biomarkers for early diagnosis of this pathology. So basic study using various models of Parkinson’s disease are very important since they may identify new targets for the treatment and new diagnostic biomarkers. The authors investigated alpha-synuclein pathogenic modifications in response to paraquat in a Drosophila model of Parkinson’s disease. The authors found that the chronic exposure of paraquat in fly expressing human alpha-synuclein is an efficient approach to reveal new mechanisms of environmental neurotoxins action on Parkinson’s pathology. This field of investigation is very important and the results presented in the manuscript will be interesting for the readers of IJMS.

We are glad that the reviewer acknowledged the importance of our study and the interest for the readers of IJMS. We would like to thank him for his work and suggestion of modifications, which have helped improving the text of the manuscript. In response to his comments, we have made all the requested modifications and included the missing references.

Abstract

Line 21: “..but that do not form fibrils or aggregates that are detected by the proteinase K assay”. The sense of this sentence is unclear, because something is missed here. Do the authors want to say: “..but that they do not form fibrils or aggregates” or “..but that these toxic forms of aSyn do not form fibrils or aggregates”. Should be clarified.

We thank the reviewer. We have updated this sentence as suggested: “These results suggest that PQ induces the accumulation of toxic soluble and misfolded forms of aSyn but that these toxic forms do not form fibrils or aggregates that are detected by the proteinase K assay.”

Introduction:

Line 28-29: “Syn is a vertebrate-specific 140 amino-acid presynaptic protein, which can acquire neurotoxic properties during aging and in synucleopathies such as PD.” The authors should correct “synucleopathies” on a more commonly used term “synucleinopathies” and cite here a review: ”Intracellular dynamics of synucleins: Here, there and everywhere. International Review of Cell Molecular Biology, 2015; 320:103-169.”

We modified the sentence according to the suggestion and included the missing reference as followed: “aSyn is a vertebrate-specific 140 amino-acid presynaptic protein, which can acquire neurotoxic properties during aging and in synucleinopathies such as PD (Surguchov 2015, PMID 26614873).”

Line 35: ”The accumulation of misfolded a-Syn…” the authors should add here a citation to a recent article:  “Cell responses to extracellular α-synuclein. Molecules 2019, 24 (2), 305, 21.

Thank you, we added the reference Surguchev et al. 2019 (PMID 30650656). 

Line 46: ”With non-human primates that most closely mimic the human pathology, mouse is the most used animal model…” The sense of the sentence is unclear. How non-human primates and mouse are combined here? May be the authors mean: ” Although non-human primates most closely mimic the human pathology, mouse is the most often used animal model”.

Thank you for this suggestion, we modified the sentence according to the suggestion: “Although non-human primates most closely mimic the human pathology, mouse is the most often used animal model”

Results:

Line 126: “the detection of aSyn aggregates is not routinely used due their difficult detection in tissues of Drosophila transgenic lines” should be corrected as follows:” the detection of aSyn aggregates is not routinely used due to difficulties in their detection in tissues of Drosophila transgenic lines”

We modified the sentence according to the suggestion as followed: “However, depending on the strength of the expression of aSyn, the detection of aSyn aggregates is not routinely reported due to issues in their detection in tissues of Drosophila transgenic lines  [20,21,25,28,29].”  

Line 133: ”We observed a more intense aSyn staining by immuno-fluorescence” This is a clumsy sentence which may be written as follows :” We observed a more intense immunofluorescent aSyn staining…”

Thank you, we modified the sentence according to the suggestions as followed: “We observed a more intense immunofluorescent aSyn staining in young flies expressing aSynWT or aSynA53T compared with older flies of the same genotype with the Syn 211 antibody (Figure 1A, D, G, J, M and N).”

Line 186: ”This result suggests that the misfolding or aggregation of aSyn A53T in old flies is responsible for its reduced detection in LB1 buffer.” This is an interesting assumption, which requires more detailed explanation. What is in favor of misfolding and what is in favor of aggregation?

We agree with the reviewer that the fundamental mechanism of protein misfolding that leads to protein aggregation and associated diseases remains somehow enigmatic.

For this this reason, we have chosen to modified the sentence as follow: “This result suggests that the progression of aSynA53T into the aggregative process in old flies is responsible for its reduced detection in LB1 buffer”.

Line 232: ”…typically between 10 and 20mM fed to adult flies” The acute dose of PQ used is not clearly described here. If the authors used solution of PQ with concentration 10 and 20 mM, what amount of PQ the flies received?  Should be explained more precisely.  

We have clarified this point as followed: “PQ exposure was used in Drosophila to study PD [20,34,35],[38],[36]. However, the acute dose of PQ used in these models (typically 10 or 20mM fed to adult flies, depending on the studies) lead to rapid loss of fly viability within 2-3 days, which precluded a fine analysis of the effect of PQ on aSyn protein”.

Figure 6C. The arrow with a-Syn is located at higher position than protein band on this figure.

Thank you, we have changed the arrow to point directly to aSyn in the figure 6C.

   Discussion

Line 367-368 – similar to comment to line 232: 0.75mM - a moderate fly toxicity, 10-20 mM acute treatment. Could the authors described more precisely what dose the flies received?

Thanks again, we have modified as followed: “We first found that mild dose of PQ (0.75mM) exhibited a moderate fly toxicity (LT50 between 15 and 19 days) compared to acute treatment (10 or 20mM depending on the studies), in which most of the flies die within 2-3 days [37,57].”

Round 2

Reviewer 1 Report

The authors have addressed all the concerns in the previous review. The revision has significantly improved the quality of the manuscript.

This paper has presented an exciting drosophila model with comprehensive comparisons of aging-induced with PQ-induced aSyn modifications using multiple biochemical assays. It adds a valuable piece for understanding synucleinopathies in Parkinson's disease.